# Lower Extremity Arterial Disease and Lumbar Spinal Stenosis: A Study of Exercise-Induced Arterial Ischemia in 5197 Patients Complaining of Claudication

**DOI:** 10.3390/jcm11195550

**Published:** 2022-09-22

**Authors:** Simon Lecoq, Jeanne Hersant, Mathieu Feuilloy, Henri-François Parent, Samir Henni, Pierre Abraham

**Affiliations:** 1Sports Medicine, University Hospital Angers, 49000 Angers, France; 2Vascular Medicine, University Hospital Angers, 49000 Angers, France; 3Superior School of Electronics ESEO, Université Catholique de l’Ouest, 49000 Angers, France; 4Clinique Saint Leonard Trélazé, 49800 Trélazé, France; 5Unité Mixte de Recherche Mitovasc, CNRS 6015-INSERM 1083, University of Angers, 49000 Angers, France

**Keywords:** diagnosis, treadmill testing, pain, buttock, calf, walking, lower extremity artery disease, osteoarthritis, spine

## Abstract

Only few studies have analyzed the associations of lower extremity artery disease (LEAD) with lumbar spinal stenosis (LSS), although it is expected to be a frequent association. With exercise-oximetry, we determined the presence of exercise-induced regional blood flow impairment (ischemia) in 5197 different patients complaining of claudication and referred for treadmill testing. We recorded height, weight, age, sex, ongoing treatments, cardiovascular risk factor (diabetes, high blood pressure, current smoking habit), and history of suspected or treated LSS and/or lower limb revascularization. An ankle-brachial index at rest < 0.90 or >1.40 on at least one side was considered indicative of the presence of LEAD (ABI+). Ischemia was defined as a minimal DROP (Limb-changes minus chest-changes from rest) value < –15 mmHg during exercise oximetry. We analyzed the clinical factors associated to the presence of exercise-induced ischemia in patients without a history of LSS, using step-by-step linear regression, and defined a score from these factors. This score was then tested in patients with a history of LSS. In 4690 patients without a history of (suspected, diagnosed, or treated) LSS, we observed that ABI+, male sex, antiplatelet treatment, BMI< 26.5 kg//m^2^, age ≤ 64 years old, and a history of lower limb arterial revascularization, were associated to the presence of ischemia. The value of the score derived from these factors was associated with the probability of exercise-induced ischemia in the 507 patients with a history of LSS. This score may help to suspect the presence of ischemia as a factor of walking impairment in patients with a history of lumbar spinal stenosis.

## 1. Introduction

Intermittent claudication is a frequent reason for medical consultation in elderly patients [1]. Arterial claudication mainly results from atherosclerotic lower extremity artery disease (LEAD), and its prevalence increases with age. Lumbar spinal stenosis (LSS) is also a degenerative disease affecting elderly patients. Both vascular and neurogenic claudication description can include pain or generate non-painful fatigue. Classically, arterial claudication is relieved at exercise cessation, while neurogenic claudication is improved with exercise cessation, as well as sitting or bending forward. Nevertheless, in routine practice, it is not always easy to distinguish between the two claudication origins. Furthermore, LEAD and LSS share many risk factors such as age and obesity [2,3,4]. The treatment of claudication resulting from either LEAD or LSS aims to improve the functional status. However, up to 25% of patients treated for LSS [5,6], and 32% of patients treated for LEAD [7] are fairly or not improved in the mid- or long-term after surgical treatment. Some of these unsatisfactory results might result from complex situations with associated neurogenic and arterial claudication. These various points underline the interest of studying the association of LSS with LEAD. To date, only a few studies have focused on LEAD and LSS in patients with claudication [4,8,9,10,11].

The most accessible routine investigation for LEAD detection is the ankle-brachial index (ABI) measurement at rest. When measurable, ABI was reported to provide from 83 to 99% specificity for the detection of LEAD while sensitivity ranges from 61 to 79% [12,13,14,15]. In addition, ABI at rest is frequently normal if lesions are not located on the measured axis (such as lesions of the internal iliac artery) [16]. Furthermore, we recently showed that 14.8 to 20.0% of patients with an abnormal ABI at rest have no exercise induced ischemia while exercise-induced ischemia can be found in 35.4% of patients with a normal ABI at rest [16,17,18].

Exercise-oximetry can be used to detect exercise-induced ischemia for patients referred for claudication of unknown, debatable, or arterial-suspected origin [15,19]. Its specific interest is to detect isolated proximal ischemia (as can result from internal iliac artery lesions), or exercise-induced hypoxemia as a cause or worsening factor of walking impairment [15,18,19]. Exercise-oximetry has been used as a routine in the last 20 years in our service for patients complaining walking limitation. The primary aim of the present study was to determine the prevalence of exercise-oximetry results showing exercise-induced ischemia in patients complaining claudication with a history of suspected, diagnosed, or treated LSS. The secondary aim was to determine in patients without a history of suspected, diagnosed, or treated LSS, which are factors that are available in primary care, could be considered for actively researching exercise-induced ischemia as a possible cause of walking impairment and to propose a score to estimate the probability of the presence of exercise-induced ischemia in patients with a history of LSS.

## 2. Materials and Methods

### 2.1. Population

Since 1999, all patients with claudication that were referred for exercise-oximetry in the Sports Medicine and Vascular Medicine services of the University Hospital of Angers had their medical file recorded in a database. As per routine practice, all patients were systematically informed that their record could be used for research purposes and were systematically informed that they had the option to deny the recording of their medical file in the database. After approval by the local ethics committee (Ref 2022-091), and according to French law, no individual consent was required from the patients to retrospectively study the database that was secured and has been administratively approved (CNIL: Commission Nationale Informatique et Liberté; reference 2017-220). Non-French speakers, persons <18 years old, adults protected by law, were not included into the database.

For this retrospective analysis, Patients that did not have exercise-oximetry results for medical or technical reasons were excluded. For patients with multiple tests, only the first visit was included.

As per routine, height and weight (to calculate body mass index: BMI), age, sex, ongoing treatments (focusing on anti-platelet, cholesterol lowering, and antihypertensive drugs), ABI at rest, cardiovascular risk factor such as diabetes mellitus (based on ongoing antidiabetic treatment), current smoking status, vascular investigations (results from ultrasound or angiography) and history of (suspected, diagnosed, or treated) LSS, as well as history of lower limb revascularization, were systematically recorded. An ABI value of < 0.90 or >1.40 at least on one side was considered indicative of the presence of LEAD and encoded ABI+ [20,21]. 

### 2.2. Treadmill Tests 

First, patients were asked to walk 10 m on a corridor at usual walking pace. The time required to walk these 10 m was used to adapt the speed of the treadmill. Patients walked on the treadmill under electrocardiographic monitoring and medical supervision. The tests were stopped due to limiting limb or non-limb symptoms, exhaustion, or for security reasons (e.g., arrhythmia, hypoxemia, abnormal ST segment depression). All treadmill tests were performed using a standard treadmill speed of 3.2 km/h with a grade of 10%. Nevertheless, the maximum speed was limited to 2.0 km/h for the patients that were not able to complete the 10 m in less than 12 s. All tests were stopped after 20 min until 2009. Since 2010, at minute 15, the constant speed and grade were changed to an incremental procedure, as previously described [22].

### 2.3. Exercise-Oximetry Recordings

All exercise-oximetry recordings were performed using Perimed^®^ PF6000 (Järfälla, Sweden) or Radiometer^®^ E5250 (Brønshøj, Denmark), with E5250 probes (Radiometer^®^, DK) positioned on the chest, on both buttocks and calves. Each test was preceded by a pre-test period of 10 to 15 min to heat the skin to 44 °C. Exercise-oximetry values were automatically correct to 37 °C and transferred in real-time to a computer. A custom-made program calculated the decrease from rest of oxygen pressure (DROP) at each lower limb site. The DROP index is defined as the difference between limb absolute value changes and chest absolute value changes from rest. A minimal DROP value (DROPmin) under –15 mmHg, during or following exercise, was considered as an argument for the presence of exercise-induced ischemia [15,18,19]. DROPmin has been validated as a sensitive and specific indicator for exercise-induced ischemia for both proximal and distal arterial claudication. A previous study showed that the treadmill procedure had little influence on exercise-oximetry results in the patients that were candidates for a slow treadmill procedure [23].

### 2.4. Score Determination and Validation

Age and BMI were dichotomized from their respective median values and male/female sex was encoded as 1/0. Thereafter, the factors associated to the presence of exercise-induced ischemia in patients without a history of LSS were identified by a step-by-step linear regression analysis. The model provided a normalized “beta” (β) value for each significant factor of the regression model. To obtain a score that could be calculated by mental calculation and would be easily memorized, the β values had to be converted to non-decimal numbers (points for the score). For this purpose, each β value was multiplied by the same coefficient (Kappa: κ) and rounded to the closer non-decimal number: Alpha (α). Addition of these α (numbers of points for the score) resulted in a score. The determination of the optimal κ coefficient to be used to convert β to α was performed in three consecutive steps. Firstly, the “smallest κ coefficient” that, when multiplied by all β values and after rounding to the closer unit, resulted in all α results being not null was determined. Then, the difference (delta: δ) between each α and κ· β product was calculated. Secondly, the deviation of the final score between the sum of κ· β products and the sum of α rounded values was calculated. Lastly, 10 different κ coefficients above the “smallest κ coefficient” observed were tested. The optimal coefficient was the one resulting in the smallest deviation. 

Finally, α values found in the patients without a history of LSS for the optimal κ coefficient were applied to patients with a history of LSS to perform an external validation of the score. In patients with a history of LSS, the proportion of tests showing exercise-induced ischemia for each observed value of the score was analyzed.

Note that additional analyses are provided online focusing on the sub-group of patients complaining buttock claudication only. The additional analyses can be found in the Appendix A.

### 2.5. Statistical Analysis

Results were expressed as mean ± standard deviation or median [25°–75° centiles] according to normal or non-normal distributions, respectively. For categorical data, results were reported as number of observations (percentage). Results of patients with and without LSS were compared using unpaired *t*-tests and Chi-square tests. Then, the performance of the score in patients with or without a history of LSS was tested with receiver operating characteristics curves (ROC). Significance of area under ROC was tested against random results. All statistical analyses were performed using SYSTAT for Windows, version 15.0.1 (SPSS Inc. IBM, Bois colombe, France). For all statistical tests, a two-tailed probability value of *p* < 0.05 was used to indicate statistical significance.

## 3. Results

Over the study period (Figure 1), 7631 exercise-oximetry tests were performed in patients that did not express opposition to the use of their medical file. Ninety-four tests were excluded for absent or incomplete exercise-oximetry results. The remaining 7537 tests resulted from 5197 different patients. After exclusion of all re-tests, 507 (9.8%) patients had a history of LSS.

Characteristics of patients with or without a history of LSS are reported in Table 1.

As shown in Table 1, patients were predominantly males and patients with a history of LSS were older than the patients without a history of LSS. Note that most patients had a history of cardiovascular disease but direct access to lower limb vascular imaging was rarely available at referral (<10% of cases). 

No complication occurred during the treadmill tests. A typical example of exercise-oximetry recording is shown in Figure 2.

The results of the treadmill tests observed in the population are reported in Table 2. As reported in Table 2, except for a shorter walking time and slightly higher median DROP values in patients with than without a history of LLS, no difference was observed between these two groups. Interestingly, proximal exercise-induced pain by history was twice more frequent in patients with than without a history of LSS. The difference persisted when the localization of pain during the treadmill test was considered.

As shown in Figure 1, and as expected, most patients with an abnormal ABI had a significant ischemia during the treadmill test. Specifically, the proportions of results showing exercise-induced ischemia were 67.8% in patients with a history of LSS and ABI+, 33.9% in patients with a history of LSS and ABI-, 82.1% in patients without a history of LSS and ABI+, and 56.7% in patients without a history of LSS and ABI- patients.

For the regression analysis of the factors associated with the presence of exercise-induced ischemia in the patients without a history of LSS (Table 3), six significant factors were identified (r = 0.389; *p* < 0.001). These factors were ABI+, male sex, antiplatelet treatment, BMI < 26.5 kg/m^2^, a history of lower limb revascularization, and age ≤ 64 years old.

The smallest κ coefficient to be used was 15, and the one resulting in the smallest deviation of rounded score from the non-rounded score was 21. After multiplying all beta values in the regression model by 21 and rounding to the closer unit, the α values in the prediction model were 5, 4, 3, 1, 1, and 1, respectively, resulting in a score ranging from 0 to 15 points. This score showed an area under the ROC curve of 0.736 ± 0.008, *p* < 0.001, for patients without a history of LSS. 

When the score defined in patients without a history of LSS was applied to the patients with a history of LSS, the probability of the presence of exercise induced ischemia was proportional to the score (Figure 3).

The proportion of patients with exercise-induced ischemia was equal or lower than 20% for scores < 5 (which implies having ABI-), while it exceeded 70% for scores > 12 (which implies being a male under antiplatelet treatment and having ABI+ plus at least one of the other three factors). The area under the ROC curve in patients with a history of LSS was 0.750 ± 0.021, *p* < 0.001 (Figure 4).

## 4. Discussion

We found in our study a 9.8% prevalence of exercise-induced ischemia in LSS+ patients. Furthermore, we observed that an abnormal ABI, male sex, antiplatelet treatment, lower limb revascularization, BMI, and age are the six factors associated with the presence of exercise-induced ischemia.

It was previously reported that approximately 47% of individuals over 60 years old will experience lumbar spinal stenosis [24]. Thereby, the 9.8% proportion of patients with a history of LSS among all patients referred to the two services in our hospital for an investigation of claudication of doubtful, complex, or vascular suspected origin appears amazingly low. This can possibly be explained by the fact that most of our patients were referred from vascular physicians that may underestimate the importance of LSS detection. Another reason is that 36% of our patients were younger than 60 years old. Another explanation can be the fact that LSS was recorded only from patient’s history and that with most patients being on ambulatory care, we did not have direct access to their medical file. As previously reported, the belief in the general population is that claudication is a pain in the calves, and many patients do not report their exercise-related proximal pain. Similarly, it is possible that some patients did not report their history of LSS when facing physicians from another specialty. We believe that it was not the case in our population because there was a very high prevalence of buttock pain by history due to a specific interest in our services in non-calf claudication. Similarly, in patients primarily investigated in a neurosurgery department, the proportion of LSS patients (older than 80 years old) who had LEAD was also amazingly low (4.5%) [25]. Indeed, the prevalence of LEAD increases with age, with a prevalence of already 12% between 70 and 74 years old in high-income countries [26]. This observation, in addition to the one reported in our study, advocate for the need of a better detection of LSS in vascular medicine structures and of better detection of LEAD in structures primarily interested in LSS.

If symptoms are not discriminants, our experience shows that in the clinical primary care, many physicians will rule out LEAD by pulse palpation, in parallel with spinal imaging. Nevertheless, pulse palpation has a good specificity (98%) but a weak sensitivity (58%) to predict a LEAD [27]. Improved sensitivity is expected from ABI measurement. The interest of ABI in LSS patients has been underlined for years [28]. The ABI remains minimally technically demanding and is available to the primary care physician. Then, although alternative methods such as post-exercise ABI, toe-brachial index, or Doppler waveform analysis are indicated, specifically in non-compressible ABI [21], they are not accessible to family primary care physicians and were not considered in the present study. Park et al. tried to determine in a retrospective study of 186 cases the factors associated with a high prevalence of PAD in LSS and intermittent claudication. They identified that male gender with diabetes and/or hypertension were the greatest risk factors for PAD in the LSS population [29]. Similarly, Uesugi et al., in a prospective analysis of 570 patients presenting with LSS, found a 6.7% prevalence of PAD. They identified advanced age, diabetes mellitus, and history of cerebrovascular disorder or ischemic heart disease as strongly associated with PAD in these LSS patients [30]. Although of major interest, these studies probably underestimated the prevalence of PAD because they used ABI as the only screening tool. Indeed, there are limitations to the use of ABI in LSS patients. Imagama et al. supported the use of ABI and TBI to detect PAD in LSS patients even if arterial pulses are normal. Interestingly, this study found the PAD occurred in 26% of subjects presenting LSS and had a particularly high frequency in those with normal ABI and abnormal TBI (22%) [31]. Furthermore, neither ABI nor TBI are sensitive to lesions toward the hypogastric circulation (such as internal iliac artery lesions). The specific interest of our study is to provide evidence of exercise-induced ischemia and not only for the presence of PAD and to diagnose proximal (buttock) ischemia. 

Obviously, ABI was the factor with the highest coefficient in our score, but the sensitivity of ABI at rest to detect exercise-induced ischemia was low in our population. This low sensitivity could be explained by the fact that we included buttock ischemia among positive results. Indeed, we previously reported the low sensitivity of ABI in this specific localization [16]. In a previous study, we also showed that 35.4% of patients with proximal claudication and normal ABI had a decrease in DROP values [16], whereas exercise-oximetry has a good sensitivity (89%) and specificity (86%) to predict significant lesions (>75%) of pelvic arteries [32]. It is notable that 2504 of our patients (48.2%) had a normal ABI at rest. This proportion is extremely high. This could appear as a bias of the study, but we believe that it only highlights the fact that most patients had atypical claudication, claudication of uncertain or doubtful origin, even when ABI was already abnormal. Lastly, in perspective of the high rate of patients with positive exercise-oximetry results in our patients with a history of LSS, it seems advisable to screen for LEAD in the management of patients with LSS. Clearly the rate of positive tests in patients with a history of LSS was higher in ABI+ than ABI- patients, but it still reached 33.9% in case of a normal resting ABI.

Our analysis of these ABI- patients with a history of LSS also suggests that a score above 6 (observed in 120 out of 277 patients) results in a probability of the presence of exercise-induced ischemia ≥ 50% and should alert the physician for a possible arterial participation to the walking impairment despite a normal ABI. The absence of “LEAD on imaging” as a predictive factor may appear unexpected. We believe that it only reflects the fact that, at referral, we had a very limited access to this information. We believe that this very low number of available results for imaging is not a limit of the study, as it reflects what would occur in primary care. Our score remains to be tested and validated in a prospective study. Similarly, we expected that limb revascularization would appear as a predictive factor because we previously showed that buttock claudication is more frequent after aorto-iliac procedures, resulting in iliac intern artery interruption and ABI possibly being normalized [33].

We advocate for the systematic detection of LEAD in patients with a history of LSS and similar systematic detection of LSS in patients with claudication and abnormal ABI. We also believe in the interest of exercise diagnostic tests, but considering that, even post-exercise ABI is possibly not the ideal candidate. Exercise-oximetry is a non-invasive technique and has proven its interest in modifying the diagnostic in case of claudication of suspected, debatable, or questionable origin [17]. Unfortunately, although it is progressively spreading in France and abroad, it is not readily accessible. Other approaches using changes in walking capacity for two consecutive walking tests could help discriminate exercise-induced pain of neurogenic or arterial origin [34]. Other non-invasive techniques, such as near infra-red spectrometry, remain to be tested.

Interestingly, more than half of the patients without a history of LSS with positive arterial results showed ischemia at the proximal level. Inversely, 18% of these patients without a history of LSS showed no ischemia at exercise despite the presence of an abnormal ABI. This suggests that the limiting factor for exercise in these patients was not ischemia, among which possibly unknown LSS and non-neurogenic causes such as osteo-articular or cardio-respiratory diseases might be present. Also notable is that even with a normal ABI at rest, one-third of the patients with a history of LSS showed a significant ischemia on treadmill, suggesting that a vascular origin was responsible for (or at least participated in) the limitation of walking capacity.

There are limitations to the present study.

First, our services are clearly not primary care facilities. However, we are dealing with ambulatory (and not hospitalized) patients, and we have limited access to each patient’s medical file. This situation mimics what happens with general practitioners in their routine practice. Second, we do not analyse the responsibility of LEAD in the pathophysiology of lumbar spinal stenosis. It has previously been discussed [35,36,37], and we only report the frequency of the association. Third, we do not report follow-up results after vascular or spine surgery. It would clearly be of interest to define whether the presence of a positive exercise-oximetry test is predictive of weak functional improvement after LSS surgery in case of an association of LSS and LEAD. This remains to be studied. Fourth, it was shown that the presence of aortic calcifications on spine imaging is associated with peripheral artery disease in a population of patients with LSS [38]. It may be interesting to include the presence of calcification as an additional factor in the risk score to detect LEAD among LSS patients. A prospective study is required to test this idea but could not be tested here because of the lack of routine access to these images. Fifth, this single-center retrospective study in a selected population will have to be externally validated. Last, it was impossible to know how many patients with no history of LSS, were later diagnosed with LSS. This would require a systematic recall of our patients, and a specific ethical approval that we do not have.

## 5. Conclusions

A history of LSS was frequent (although probably still underestimated) in patients referred in our departments of sports and vascular medicine for performing exercise oximetry. In the patients with a history of LSS, we propose a simple score derived from patients without a history of LSS that may help to detect which patient is likely to have exercise-induced ischemia and which should be referred for secondary care investigations. Beyond the score calculation, as a rule of thumb, in patients with a history of LSS, it is likely that males under antiplatelet treatment, with an abnormal ABI, and either BMI < 26.5 kg/m^2^, a history of lower limb revascularization, or an age ≤ 64 years should be investigated for an arterial participation of their walking impairment before spine surgery (because 70% will show exercise-induced ischemia). Future prospective studies are needed to confirm this suggestion.

## Figures and Tables

**Figure 1 jcm-11-05550-f001:**
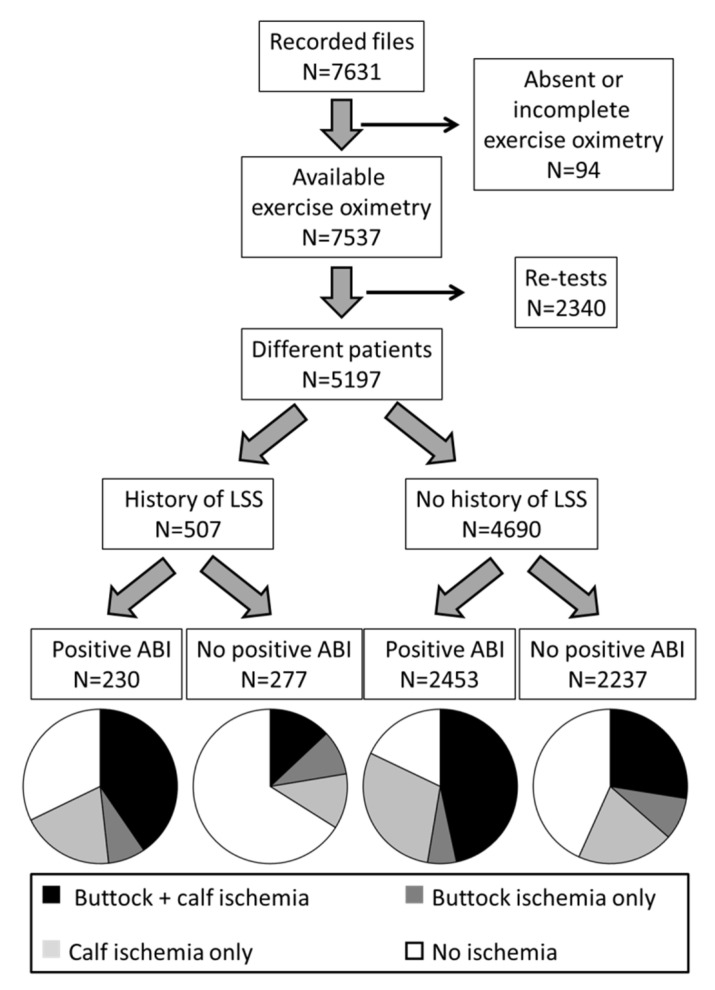
Flowchart of the study and distribution of the presence/absence of a history of suspected, diagnosed, or treated lumbar spinal stenosis (LSS), an abnormal (positive) ankle-brachial index (ABI) defined as ABI < 0.90 or >1.40. The presence and localization of exercise-induced ischemia on exercise-oximetry (TcpO2) are reported for each group of patients using circle graphs.

**Figure 2 jcm-11-05550-f002:**
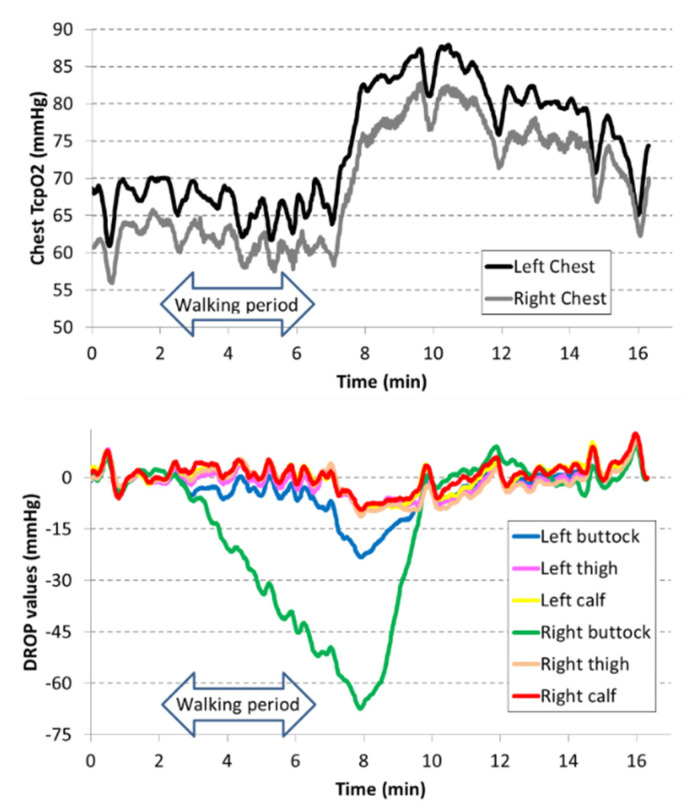
Typical example of exercise-oximetry recording expressed as DROP index in an 80-year-old patient that had a laminectomy for LSS 20 years ago and is treated by prednisone for a chronic inflammatory disease. He also had aorto-bi-iliac revascularization. ABI was 0.77 on the right side and 0.83 on the left side due to bilateral occlusion of femoral arteries. The patient complained of exercise-induced intense right buttock pain and moderate left buttock discomfort during walking. In this patient, a second chest measurement and bilateral thigh measurements were recorded, in addition to chest, buttocks, and calves. As shown, walking induces a dramatic decrease in right buttock DROP value, as well as a moderate decrease in the left buttock. Despite the bilateral occlusion of femoral arteries, no ischemia was observed in the calves during or following the walking period. Note that, despite different starting values, the changes in the two chest reference electrodes were similar. This highlights the interest in using the DROP index (which is insensitive to absolute starting values).

**Figure 3 jcm-11-05550-f003:**
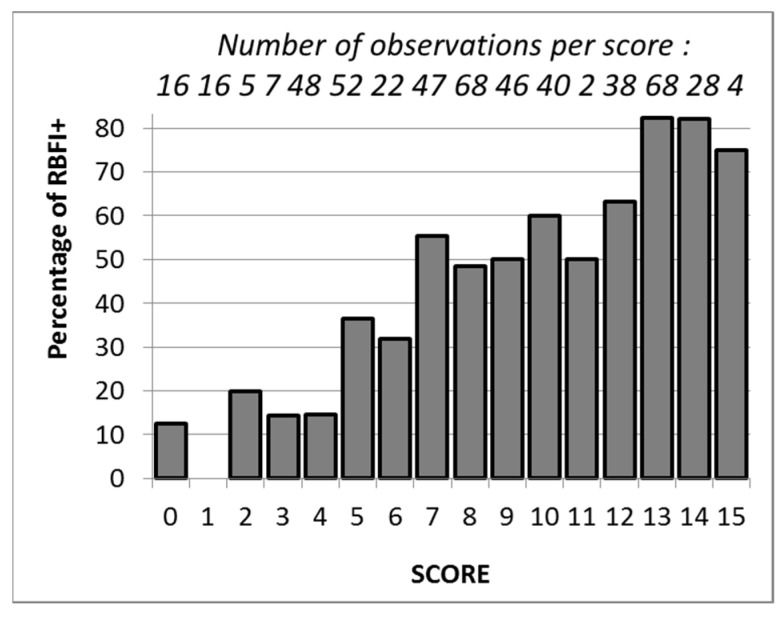
Proportion of patients with a history of lumbar spinal stenosis showing an exercise-induced ischemia as a function of the score defined in Table 3.

**Figure 4 jcm-11-05550-f004:**
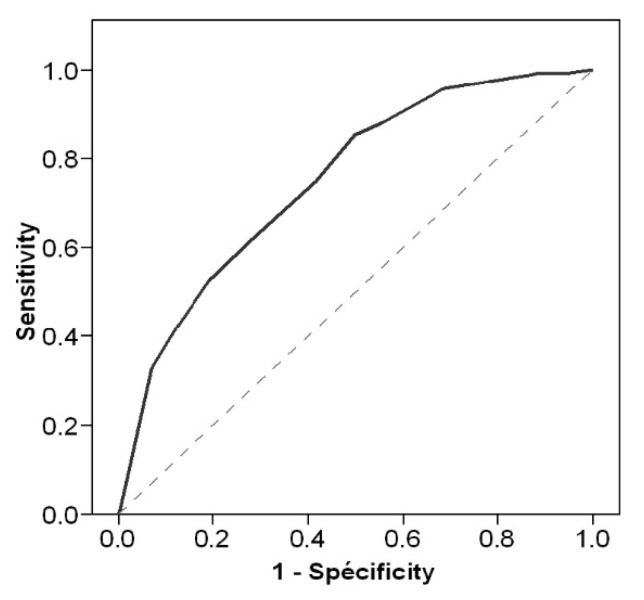
Receiver operating characteristics curves for the score to predict the presence of exercise-induced ischemia (exercise induced ischemia) in patients with a history of lumbar spinal stenosis.

**Table 1 jcm-11-05550-t001:** Characteristics of the studied patients with or without a history of suspected, diagnosed, or treated lumbar spinal stenosis (LSS). LEAD stands for lower extremity artery disease, and ABI stands for ankle-brachial index.

	History of LSS n = 507	No History of LSS n = 4690	*p*
Male sex	402 (79.3)	3688 (78.6)	0.732
Weight (kg)	80.9 ± 15.6	76.0 ± 15.3	0.001
Height (cm)	169 ± 9	168 ± 8	0.001
Age (years old)	67.5 ± 10.3	63.2 ± 11.9	0.001
Antiplatelet agent	316 (62.3)	3330 (71.0)	0.001
Antihypertensive drugs	314 (61.9)	2606 (55.6)	0.006
Cholesterol lowering drugs	280 (55.2)	2701 (57.6)	0.307
Active smokers	79 (28.0)	702 (30.9)	0.713
Pain by history on left buttock	281 (55.4)	1300 (27.7)	0.001
Pain by history on right buttock	281 (55.4)	1318 (28.1)	0.001
Pain by history on left thigh	136 (26.8)	769 (16.4)	0.001
Pain by history on right thigh	134 (26.4)	758 (16.2)	0.001
Pain by history on left calf	262 (51.7)	2430 (51.8)	0.954
Pain by history on right calf	256 (50.5)	2536 (54.1)	0.128
Time over 10 m (s)	10.4 ± 2.7	9.7 ± 2.2	0.001
History of cardiovascular disease	230 (45.4)	3061 (65.6)	0.001
History of lower limb revascularization	78 (23.3)	682 (25.5)	0.610
LEAD according to ABI (<0.90 or >1.40)	230 (45.4)	2237 (47.7)	0.318
Right positive ABI	179 (35.3)	1780 (38.0)	0.243
Left positive ABI	165 (32.5)	1907 (40.7)	0.001

**Table 2 jcm-11-05550-t002:** Results of the treadmill test with exercise-oximetry in patients with (n = 507) or without (n = 4690) a history of suspected, diagnosed, or treated lumbar spinal stenosis. Exercise-oximetry results are expressed in absolute values (TcpO2) and in lowest decrease from rest of oxygen (DROPmin) index.

	History of LSS	No History of LSS	*p*
Maximal walking time (sec)	291 [170; 563]	302 [180; 678]	0.007
Heart rate at rest (beats/min)	80 ± 14	80 ± 15	0.315
Heart rate at end exercise (beats/min)	119 ± 22	120 ± 23	0.104
Chest TcpO2 at rest (mm Hg)	67 ± 12	67 ± 13	0.932
Minimal chest TcpO2 (mm Hg)	62 ± 12	62 ± 13	0.532
Left buttock TcpO2 at rest (mm Hg)	68 ± 11	69 ± 13	0.281
Left calf TcpO2 at rest (mm Hg)	70 ± 10	71 ± 12	0.031
Right buttock TcpO2 at rest (mm Hg)	68 ± 11	69 ± 12	0.216
Right calf TcpO2 at rest (mm Hg)	70 ±11	71 ± 13	0.128
Left buttock DROPmin (mm Hg)	−9 [−16; −5]	−11 [−20; −6]	0.001
Left calf DROPmin (mm Hg)	−11 [−19; −6]	−15 [−26; −8]	0.001
Right buttock DROPmin (mm Hg)	−8 [−15; −5]	−11 [−19; −6]	0.001
Right calf DROPmin (mm Hg)	−10 [−17; −6]	−14 [−27; −8]	0.001
DROPmin < −15 mmHg on one or both buttocks	173 (34.2)	2071(44.2)	0.001
DROPmin < −15 mmHg on one or both calves	206 (40.6)	2874 (61.3)	0.001
Left buttock pain on treadmill	214 (42.2)	1199 (25.6)	0.001
Right buttock pain on treadmill	206 (40.6)	1208 (25.8)	0.001
Left thigh pain on treadmill	49 (9.7)	309 (6.6)	0.010
Right thigh pain on treadmill	48 (9.5)	322 (6.9)	0.020
Left calf pain on treadmill	239 (47.1)	2464 (52.5)	0.021
Right calf pain on treadmill	235 (46.4)	2428 (51.8)	0.020

**Table 3 jcm-11-05550-t003:** Results from the step-by-step linear regression analysis in patients without lumbar spinal stenosis of factors predictive of the presence of exercise-induced ischemia. The points for the score are the alpha non-decimal values obtained as explained in the text.

Studied Parameters	Beta	SE	Normalised Beta	*p*	Points for the Score
ABI+	0.219	0.013	0.237	<0.001	+5
Male sex	0.217	0.015	0.192	<0.001	+4
Antiplatelet treatment	0.168	0.014	0.164	<0.001	+3
BMI < 26.5 kg/m^2^	0.056	0.013	0.061	<0.001	+1
Lower limb revascularization	0.059	0.019	0.045	<0.01	+1
Age ≤ 64 years old	0.033	0.013	0.035	0.01	+1
Antihypertensive drugs	0.020	1.391		0.867	-
Cholesterol lowering treatment	0.003	0.172		0.808	-
Diabetes mellitus	0.017	1.200		0.944	-
Active smoking	0.000	0.005		0.959	-

## Data Availability

The data underlying the present study may be available upon reasonable request to the authors.

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
