# Peer review of "Lower Extremity Arterial Disease and Lumbar Spinal Stenosis: A Study of Exercise-Induced Arterial Ischemia in 5197 Patients Complaining of Claudication"

_jcm, 2022, doi:10.3390/jcm11195550_

Round 1
Reviewer 1 Report
The study aims to highlight the coexistence of LSS and LEAD and furthermore, to identify the risk factors in patients with LSS suggesting the importance of investigation for an arterial origin. The strength of the study is an inclusion of over 5000 patients, also it has been done in a retrospective manner. In addition, an exercise oximetry was conducted for all patients, which could specifically recognize a disturbance in perfusion. Based on previous studies, a co-existence of LSS and LEAD is often observed, triggering a discussion about the indication for further arterial survey for patients with LSS before surgical treatments. Considering the unsatisfying improvement in these patients with LSS after surgery, the clinical importance is beyond question. However, I have some questions:
Abstract:
- The abstract is hard to follow. A more structured abstract would be easier to read.
Introduction:
- The introduction is very long, it can be shortened in my eyes
Material and Methods:
- Was ethics approvement needed?
- LEAD is defined as a ABI value <0.9 or > 1.4. However, according to 2017 ESC Guidelines on the Diagnosis and Treatment of Peripheral Arterial Diseases, in collaboration with the European Society for Vascular Surgery (ESVS), alternative methods such as the toe-brachial index or Doppler waveform analysis are indicated. Were there further investigations performed/recorded?
- Was your ABI recorded at rest or after provocation?
- The definition for LSS+ patients is not clear
- How were the results of treadmill tests reported? Information about the pain characteristics and locations would be of importance in this context, beside only absolute/relative walking distance.
Results:
- Would it make sense to analyze patients with buttock claudication and calve claudication separately? In my eyes the overlapping of diagnosis is mostly relevant in LSS and buttock claudication. Therefore, a score to detect selectively these patients would really make sense. However, currently your score is also influenced by calf claudicants, that normally present with different symptoms.
Conclusion:
- The conclusion can be shortened and focused on the relevant finding: Positive score leads to further vascular investigation
General comments:
The general manuscript is very hard to follow, as there are many abbreviations and the statistic is very complex. Maybe it would make sense to skip out information that are not relevant for the reader. Furthermore, the manuscript would benefit from English editing.
Author Response
Please find our responses in the attached file

Reviewer 2 Report
1)please clarify as to whether TcpO2 correlate to area of symptoms
2)Page 6, last paragraph indicated 18% of LSS- patients showed no ischemia at exercise despite the presence of an abnormal ABI, does this mean the pain is related to a non-neurogenic and non-vascular causes? please elaborate.
Author Response

(The authors gave the same response as above.)

Reviewer 3 Report
In the present manuscript, the authors investigated the association between lower extremity arterial disease (LEAD) and lumber spinal stenosis (LSS). This study defined lower extremity ischemia as exercise-induced regional blood flow impairment (DROP value < -15 mmHg) with oximetry (tcpO2). In patients with a history of suspected, diagnosed or treated LSS, lower extremity ischemia was induced in 67.8% of patients with ABI abnormal and 33.9% of patients with normal ABI, suggesting a high prevalence of LEAD combined with LSS. The authors also developed a scoring system to predict lower extremity ischemia in patients without a history of LSS. This predictive score, which included abnormal ABI, male, antiplatelet medications, low BMI, history of lower extremity revascularization, and young age, showed similar predictive value in the group of patients with a history of LSS. A score of 7 or higher predicted lower extremity ischemia in more than 50% of patients.
This study is novel in that it demonstrates lower extremity ischemia not by ABI, but by tcpO2, which can evaluate it in more detail, and is interesting because it shows the relationship of LEAD to LSS.
1. Add a note to the Abstract regarding the frequency of lower extremity ischemia in patients with a history of LSS.
2. The description in the Abstract “The score derived from these factors was strongly associated to the probability of RBFI+ results in the 507 LSS+ patients.” (in the line 23–24) is inappropriate. The specific predictive ability of the score should be stated.
3. In the Methods section and Figure 1, there is no description of the group of patients for whom tcpO2 was indicated. Add a statement that patients with claudication were evaluated.
4. Interpretation and discussion of data should be done in the Discussion section, not in the Results section. (in the line 203–210)
5. Lines 211 "LSS+" should be "LSS-". To be confirmed.
6. Provide a predictive cutoff value for lower extremity ischemia in LSS+ patients and its sensitivity and specificity, as they are important in clinical practice and for comparison with other studies.
7. Summarize the results of this study in the first paragraph of the Discussion and clearly describe the answers to the purposes mentioned in the Introduction.
8. There appears to be a duplicate citation in the text (citation [31] in the line 270 and 273). To be confirmed.
9. This is a single-center retrospective study and, as noted in the Discussion section, included a selected patient population. In addition, the performance of this score was not validated in other populations. These should be described in the Limitation paragraph.
10. The Conclusion section should be more concise. As mentioned above, a summary of this study should be provided in the first paragraph of the Discussion.
Author Response

(The authors gave the same response as above.)

Round 2
Reviewer 1 Report
Thank you for the chance to read the modified version again.
By adding the separate analysis the manuscript significantly improved.
Reviewer 3 Report
I have reviewed the revised manuscript.
No further comments.